# Decoupling Sign and Magnitude in Optimization with Adaptive Momentum Scaling

## Abstract

We introduce Adaptive Momentum Scaling (AMS), a general optimization framework that decouples the direction and magnitude of parameter updates by separately tracking the sign and scale of momentum. AMS unifies and extends existing optimizers; in particular, we show that it recovers Adam and Cautious Adam as special cases through appropriate hyperparameter choices. Building on this framework, we develop Gradient Descent with Adaptive Momentum Scaling (Grams), a novel optimizer that leverages gradient direction for updates while using momentum exclusively for adaptive magnitude scaling. This design enables Grams to achieve more effective loss descent than conventional momentum-based and cautious methods. We provide theoretical guarantees for Grams, including discrete-time descent analysis , and further connect its dynamics to Hamiltonian descent. Empirically, Grams consistently outperforms widely-used optimizers such as Adam, Lion, and their cautious variants across a range of tasks, including pre-training and fine-tuning large language models. Our results demonstrate that AMS and Grams offers a principled and scalable solution for modern deep learning optimization.

## 1 Introduction

Optimization plays a pivotal role in modern machine learning, serving as the cornerstone for training and fine-tuning models across diverse applications. Over the past decade, the introduction of adaptive optimizers like Adam (Kingma & Ba, 2014) and its variant AdamW (Loshchilov & Hutter, 2017) has significantly shaped the landscape of optimization. These algorithms have become the de facto choices for a variety of tasks, ranging from pre-training Large Language Models (LLMs) (Touvron et al., 2023) to fine-tuning models for text-to-image diffusion (Rombach et al., 2022). Despite the advent of new methods, AdamW has maintained its dominance, particularly in large-scale training regimes, thanks to its robust convergence properties and general applicability.

The era of LLMs has ushered in unprecedented scaling of model sizes, demanding billions or even trillions of parameters (Achiam et al., 2023). This scaling places an immense burden on computational resources, intensifying the need for efficient optimization strategies. A faster optimizer directly translates to the ability to process more training tokens within a fixed time budget, leading to the development of more capable models (Kaplan et al., 2020). This necessity has rekindled interest in identifying optimizers that can surpass AdamW in terms of speed, memory efficiency, and convergence guarantees.

Recent innovations, such as SHAMPOO (Gupta et al., 2018), Schedule Free (Defazio et al., 2024), Lion (Chen et al., 2024), SOAP (Vyas et al., 2024), and ADOPT (Taniguchi et al., 2024), have pushed the boundaries of optimization by introducing novel update rules, momentum mechanisms, and regularization techniques. These methods promise substantial improvements in training efficiency and model performance, particularly in specialized scenarios. The cautious (Liang et al., 2024) mechanism addresses optimization challenges by adaptively masking the momentum term $u_t$ to align with the gradient $g_t$, preventing conflicts that hinder training. This approach extends to Adam and Lion, resulting in variants like Cautious Adam (C-Adam) and Cautious Lion (C-Lion).

In this paper, we propose Gradient Descent with Adaptive Momentum Scaling (Grams), a novel optimization algorithm designed to address the limitations of existing methods. Unlike traditional optimizers that directly couple momentum with gradient updates, Grams decouples the direction

and magnitude of parameter updates. This approach allows the update direction to be derived solely from current gradients while momentum is utilized to scale the update magnitude. Such decoupling enhances stability and robustness, particularly in dynamic optimization landscapes.

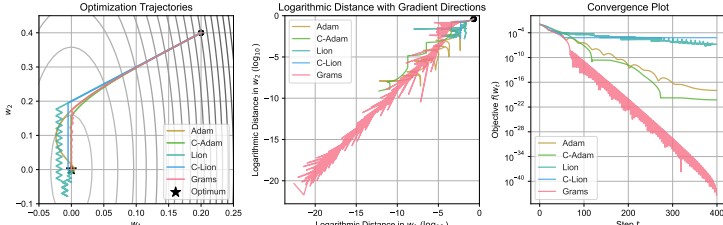

Figure 1: Convergence comparison on a simple convex function $f(w) := (0.5w_1)^2 + (0.1w_2)^2$. Learning rate $\eta = 0.01$ for Grams, Adam, and C-Adam, and $\eta = 0.001$ for Lion and C-Lion. $\beta_1$ and $\beta_2$ are default values for all optimizers. The graph on the left is the optimizing trajectories; the graph in the middle graph is the distance between current weight and optimum weight; the graph on the right is the training objectives.

Figure 1 illustrates the superior convergence properties of Grams compared to other state-of-the-art optimizers on a simple convex function. In the left graph, the optimization trajectories of Grams exhibit a combination of characteristics observed in Lion and C-Adam. Specifically, Grams follows a shortcut-like path similar to C-Adam while also demonstrating a zigzagging behavior reminiscent of Lion. However, unlike Lion, which deviates significantly from the optimal path due to its pronounced zigzagging updates, Grams maintains a more controlled trajectory, effectively balancing stability and efficiency during optimization. The middle graph illustrates the logarithmic distance of the weights $w_1$ and $w_2$ from the optimum. Here, Grams consistently demonstrates a faster descent compared to other optimizers, indicating superior efficiency in reducing the distance to the optimal solution. The right graph displays the convergence of the objective function value over training steps, where Grams achieves a notably faster reduction and lower final objective value than its competitors. These results collectively underscore Grams' ability to navigate the optimization landscape effectively, outperforming traditional and Cautious optimizers in terms of speed and precision, even in a simple convex setting.

Our contributions are summarized as follows:

- We introduce Adaptive Momentum Scaling (AMS), which is a general framework that tracks the sign and magnitude of the momentum. We proved that it can generalize Adam and Cautious Adam by tuning hyperparameters.

- We introduce the Grams optimizer, which empirically outperforms existing methods such as Adam (Loshchilov & Hutter, 2017), Lion (Chen et al., 2024), and their Cautious version (Liang et al., 2024).

- We establish theoretical guarantees for Grams, including discrete-time descent analysis and Hamiltonian descent analysis.

By integrating insights from momentum-based methods, adaptive optimizers, and sign-based updates, Grams bridges the gap between theoretical rigor and practical performance, offering a promising direction for scalable and efficient optimization in modern machine learning.

**Roadmap.** In Section 2, we review related work and place our approach in the context of existing optimization methods. In Section 3, we introduce our notation system and outline key preliminary concepts necessary for understanding our method. In Section 4, we present a general framework that tracks the sign and magnitude of the momentum. In Section 5, we present our main contribution, Gradient Descent with Adaptive Momentum Scaling (Grams), and provide theoretical guarantees for its performance. In Section 6, we evaluate the effectiveness of Grams through empirical experiments on both pre-training and fine-tuning tasks, comparing its performance to state-of-the-art optimizers. In Section 7, we conclude the paper and discuss future directions to enhance the capabilities of Grams further.

## 2 RELATED WORK

**Adam Variants and Memory-Efficient Optimization**  Adam and its numerous variants have been pivotal in addressing optimization challenges across diverse applications (Kingma & Ba, 2014; Liu et al., 2019). Among these, AdamW (Liu et al., 2019) introduced a crucial modification by decoupling weight decay from gradient updates, restoring the original intent of weight regularization. NAdam (Dozat, 2016) integrated Nesterov momentum, and AdaBelief (Zhuang et al., 2020) refined the second moment estimation for improved generalization. Adan (Xie et al., 2024) extended these advancements with an additional momentum term, balancing performance with memory overhead. Schedule-free optimizers (Defazio et al., 2024) have further simplified the optimization process by dynamically adjusting learning rates without pre-defined schedules, enhancing adaptability across tasks. More recent efforts, such as ADOPT (Taniguchi et al., 2024), streamlined first-order momentum updates through normalization.

Memory-efficient strategies have addressed the growing resource demands of large-scale models. AdaFactor (Shazeer & Stern, 2018) factorize second-order statistics, achieving sublinear memory usage. K-Fac (Martens & Grosse, 2015) approximates the Fisher information matrix using Kronecker-factored representations. Innovations such as fused gradient computation (Lv et al., 2023) and GaLore (Zhao et al., 2024) leverage low-rank gradient structures to optimize memory efficiency.

**Regularization Techniques**  Regularization plays a critical role in improving generalization and robustness in optimization. Lion (Chen et al., 2024) introduced sign-based updates with uniform magnitudes, offering inherent noise regularization (Neelakantan et al., 2017; Foret et al., 2021; Chen et al., 2022). Earlier methods, such as signSGD (Bernstein et al., 2018), explored similar ideas but focused on reducing communication costs in distributed optimization. Despite its efficiency, signSGD often underperformed in deep learning tasks, such as ConvNet training, where Lion demonstrated superior performance through advanced momentum mechanisms.

Building on these ideas, the Cautious mechanism (Liang et al., 2024) adaptively masks momentum terms to ensure alignment with gradient directions, mitigating conflicts. This approach has led to new variants, including Cautious Adam (C-Adam) and Cautious Lion (C-Lion), which combine regularization benefits with robust convergence guarantees.

**Hamiltonian Dynamics in Optimization**  Hamiltonian dynamics provides a robust theoretical framework for understanding momentum-based optimization (Nesterov, 1983; Sutskever et al., 2013; Nguyen et al., 2024; Anonymous, 2024). The seminal work of (Sutskever et al., 2013) provided a physical interpretation of momentum methods, linking the oscillatory behavior of algorithms like Nesterov's and Polyak's methods (Nesterov, 1983) to principles of dynamical systems. While traditional gradient descent guarantees a monotonic decrease in objective function values, momentum-based methods exhibit non-monotonic dynamics that require more advanced analytical tools (Jin et al., 2018). This has motivated the development of Lyapunov-based approaches for convergence analysis in convex optimization (Krichene et al., 2015; Wilson et al., 2016).

Recent studies have further formalized these connections by modeling optimization processes as continuous-time ODEs, uncovering inherent Hamiltonian structures (Maddison et al., 2018; Nguyen et al., 2024). These insights have significantly enhanced the theoretical understanding of classical momentum-based algorithms and provided a foundation for exploring new optimization frameworks (Anonymous, 2024). Moreover, Hamiltonian principles have been extended to analyze convergence rates for accelerated methods (Jin et al., 2018) and have inspired broader applications in optimization. In parallel, Mirror Descent, while distinct from Hamiltonian dynamics, leverages variational principles and maintains efficiency with a mild dependence on the dimensionality of decision variables, making it well-suited for large-scale problems (Krichene et al., 2015; Tzen et al., 2023).

## 3 PRELIMINARIES

In this section, we outline foundational concepts and notations that will be referenced throughout the paper. In Section 3.1, we define some useful notations. In Section 3.4, we summarize

the Hamiltonian dynamics framework, which provides a theoretical foundation for understanding momentum-based optimization algorithms.

## 3.1 NOTATIONS

For two vectors $u, v \in \mathbb{R}^d$, we use $\langle u, v \rangle$ to denote the standard inner product in the Euclidean space. We use $\|u\|_2$ to denote the $\ell_2$-norm of $u$ and use $\|u\|_\infty$ to denote the $\ell_\infty$-norm of $u$. For a matrix $A$, we use $\|A\|_F$ to denote the Frobenius norm of $A$. For a twice differentiable function $f : \mathbb{R}^d \to \mathbb{R}$, we use $\nabla f(x)$ and $\nabla^2 f(x)$ to denote the gradient and Hessian of $f$, respectively. Given a vector $x \in \mathbb{R}^d$, we use $\mathbf{1}_{x \geq 0} \in \mathbb{R}^d$ to denote the vector where each entry indicates whether the corresponding entry of $x$ is non-negative, i.e., for each $i \in [d]$, $(\mathbf{1}_{x \geq 0})_i = 1$ if $x_i \geq 0$, and $(\mathbf{1}_{x \geq 0})_i = 0$ otherwise.

## 3.2 SIGN FUNCTION

We formally define the sign function, which will be used later in our optimizer Grams.

**Definition 3.1** (Sign function). *Given a vector $a = (a_1, a_2, \ldots, a_n) \in \mathbb{R}^n$, the sign function of $a$, denoted as $\mathrm{sign}(a)$, is defined component-wise as:*

$$\mathrm{sign}(a) = (\mathrm{sign}(a_1), \mathrm{sign}(a_2), \ldots, \mathrm{sign}(a_n)),$$

*where the scalar sign function $\mathrm{sign}(a_i)$ is given by:*

$$\mathrm{sign}(a_i) = \begin{cases} 1, & \text{if } a_i > 0, \\ 0, & \text{if } a_i = 0, \\ -1, & \text{if } a_i < 0. \end{cases}$$

## 3.3 CAUTIOUS OPTIMIZERS

Cautious mechanism (Liang et al., 2024) addresses a key challenge in optimization dynamics: when the momentum term $u_t$ moves in a different direction from the current gradient $g_t$, it can potentially impede training progress. To mitigate this issue, the Cautious mechanism introduces an adaptive masking mechanism that modifies the momentum term based on its alignment with the gradient direction. Cautious mechanism could apply to Adam and Lion, which form Cautious Adam (C-Adam) and Cautious Lion (C-Lion).

**Definition 3.2** (Cautious Mechanism Parameter Update). *The general parameter update rule for the Cautious mechanism is given by:*

$$\widehat{u}_t := u_t \circ \mathbf{1}_{u_t \circ g_t \geq 0}$$
$$\widehat{\eta} := \frac{d}{\|\widehat{u}_t\|_1}$$
$$w_t := w_{t-1} - \widehat{\eta}\widehat{u}_t \tag{1}$$

*where $w_t$ is the weight at time step $t$, $\circ$ denotes Hadamard product. For C-Adam, $u_t$ is from Definition A.4; For C-Lion, $u_t$ is from Definition A.5. $g_t$ is the current gradient.*

The Cautious mechanism in Definition 3.2 modifies the parameter updates to ensure they align with the gradient direction, thereby reducing the risk of adverse updates that could impede convergence. To analyze the impact of this mechanism, we introduce Definition 3.3, which quantifies the change in the loss function after an update.

**Definition 3.3.** *For any loss function $\mathcal{L} : \mathbb{R}^d \to \mathbb{R}$, we define*

$$\Delta \mathcal{L}_{w_{t+1}, w_t} := \mathcal{L}(w_{t+1}) - \mathcal{L}(w_t),$$

*where $w_{t+1}$ is updated from any update rule.*

As shown in (Liang et al., 2024), the Cautious mechanism ensures that the updated parameters result in a non-negative inner product with the gradient, leading to a monotonic decrease in the loss function when the step size is sufficiently small. Specifically, using a Taylor approximation, it can

be expressed as: $\Delta\mathcal{L}_{w_{t+1},w_t} \approx -\eta_t(u_t \circ g_t)^\top \phi(u_t \circ g_t) \leq 0$, where $\phi(\cdot)$ represents the alignment mask introduced by the Cautious mechanism. This guarantees that $\mathcal{L}(w_{t+1}) \leq \mathcal{L}(w_t)$, ensuring a decrease in loss.

We formalize that the expected decrease in loss when updating the parameter $w$ from optimization step $t$ to step $t+1$ can be approximated using a first-order Taylor expansion, which indicates the loss function will decrease monotonically when the step size is sufficiently small.

**Lemma 3.4** (Informal version of Lemma D.1). *Suppose that $\mathcal{L} : \mathbb{R}^d \to \mathbb{R}$ is L-smooth. Let $\Delta\mathcal{L}_{w_{t+1}^C,w_t}$ be defined in Definition 3.3, $w_{t+1}^C$ is updated from $w_t$ using Definition 3.2. Then we have the followings:*

- *Part 1. It holds that*

$$\Delta\mathcal{L}_{w_{t+1}^C,w_t} \leq -\eta_t\langle u_t \circ g_t, \mathbf{1}_{u_t \circ g_t \geq 0}\rangle + \frac{L\eta_t^2}{2}\|u_t\|_2^2, \tag{2}$$

- *Part 2. It holds that $\Delta\mathcal{L}_{w_{t+1}^C,w_t} \geq -\eta_t\langle u_t \circ g_t, \mathbf{1}_{u_t \circ g_t \geq 0}\rangle$.*

- *Part 3. If $\eta_t \leq \frac{2}{L\|u_t\|_2^2}\langle u_t \circ g_t, \mathbf{1}_{u_t \circ g_t \geq 0}\rangle$, then $\Delta\mathcal{L}_{w_{t+1}^C,w_t} \leq 0$.*

Building on these findings, Theorem E.2 delves into the Hamiltonian properties of the Cautious mechanism, providing deeper insights into its theoretical guarantees within continuous optimization dynamics.

### 3.4 HAMILTONIAN DESCENT

Hamiltonian descent provides a theoretical framework for analyzing momentum-based optimization algorithms by introducing an augmented objective function, the Hamiltonian. This framework allows us to study optimization dynamics through the lens of continuous-time differential equations, linking the monotonic descent of the Hamiltonian function to the stability and convergence of the optimization process. We formalize this concept as Definition E.1, based on the formulation presented in Section 2.1 of (Liang et al., 2024).

## 4 ADAPTIVE MOMENTUM SCALING

We propose Adaptive Momentum Scaling (AMS), a general framework that tracks the sign and magnitude of the momentum separately. See Algorithm 1 for definition.

We then show that for some specific hyperparameters, AMS is equivalent to Adam. Formally, we have the following corollary.

**Corollary 4.1** (Informal version of Corollary B.1). *By choosing $\beta_3 = \beta_1$ or $\lambda = 1$, AMD is equivalent with Adam.*

Then, we show that we can also let AMS equivalent to Cautious Adam by set hyperparameters.

**Corollary 4.2** (Informal version of Corollary B.2). *By choosing $\beta_3 = 0$ and $\lambda = 0$, AMS is equivalent with Cautious Adam.*

## 5 GRADIENT DESCENT WITH ADAPTIVE MOMENTUM SCALING

A special case of AMS, where $\beta_3 = 0$ and $\lambda = -1$, we name it *Gradient Descent with Adaptive Momentum Scaling*, or simply *Grams*. This section formalizes the update rule of Grams, introduces its key components, and provides theoretical guarantees in both loss descent and Hamiltonian dynamics for its performance.

### 5.1 DEFINITIONS

We define the Grams optimizer formally in Definition 5.1.

---

**Algorithm 1** Adaptive Momentum Scaling (AMS)

---

**Require:** parameter $w$, step sizes $\{\eta_t\}$, dampening factors $\beta_1, \beta_2, \beta_3 \in [0, 1)$, $\epsilon > 0$, weight decay $\gamma \geq 0$, negative momentum scaling factor $\lambda \in \mathbb{R}$.

1: Initialize $t = 0, m_0 = v_0 = s_0 = \mathbf{0}$
2: **while** $w_t$ not converged **do**
3:      $t \leftarrow t + 1$
4:      $g_t \leftarrow \nabla_w \mathcal{L}_t(w_{t-1})$
5:      $m_t \leftarrow \beta_1 m_{t-1} + (1 - \beta_1) g_t$
6:      $v_t \leftarrow \beta_2 v_{t-1} + (1 - \beta_2) g_t^2$
7:      $s_t \leftarrow \beta_3 s_{t-1} + (1 - \beta_3) g_t$                          $\triangleright$ Track the sign
8:      $\widehat{m}_t \leftarrow m_t / (1 - \beta_1^t)$
9:      $\widehat{v}_t \leftarrow v_t / (1 - \beta_2^t)$
10:     $u_t \leftarrow \widehat{m}_t / (\sqrt{\widehat{v}_t} + \epsilon)$
11:     $\phi_t \leftarrow \mathbf{1}_{u_t \circ s_t \geq 0}$                                 $\triangleright$ Mask for same signs
12:     $\psi_t \leftarrow \lambda \cdot \mathbf{1}_{u_t \circ s_t < 0}$                 $\triangleright$ Mask and scale for different signs
13:     $\widehat{\eta} \leftarrow \eta \frac{d}{\|\phi_t\|_1 + \||\psi_t\|_1|}$                               $\triangleright$ Scale $\eta$
14:     $\widehat{u}_t \leftarrow (\phi_t + \psi_t) \circ u_t$
15:     $w_t \leftarrow w_{t-1} - \widehat{\eta}_t \widehat{u}_t$
16:     $w_t \leftarrow w_t - \widehat{\eta}_t \gamma w_t$                                   $\triangleright$ Add weight decay
17: **end while**

---

**Definition 5.1** (Grams Optimizer). *Grams optimizer is a special case of AMS (Algorithm 1) by set $\beta_3 = 0$ and $\lambda = -1$.*

**Corollary 5.2** (Informal version of Corollary B.3). *By choosing $\beta_3 = 0$ and $\lambda = -1$, the update rule of AMS is equivalent with: Part 1. $m_t := \beta_1 m_{t-1} + (1 - \beta_1) g_t$, Part 2. $v_t := \beta_2 v_{t-1} + (1 - \beta_2) g_t^2$, Part 3. $\widehat{m}_t := \frac{m_t}{1 - \beta_1^t}$, Part 4. $\widehat{v}_t := \frac{v_t}{1 - \beta_2^t}$, Part 5. $u_t := \frac{\widehat{m}_t}{\sqrt{\widehat{v}_t} + \epsilon}$, Part 6. $\widehat{u}_t := \text{sign}(g_t) \circ |u_t|$, Part 7. $w_t := w_{t-1} - \eta_t \widehat{u}_t$.*

### 5.2 Loss Descent

In this subsection, we analyze the loss descent properties of the Grams algorithm. Understanding how the loss function decreases over optimization steps provides insights into the efficiency and stability of the method. Below, we formalize the relationship between the step size, gradients, and the resulting decrease in the loss value, leveraging the $L$-smoothness property of the objective function.

**Lemma 5.3** (Informal version of Lemma D.2). *Suppose that $\mathcal{L} : \mathbb{R}^d \to \mathbb{R}$ is $L$-smooth. Let $\Delta \mathcal{L}_{w_{t+1}^{\text{Grams}}, w_t}$ be defined in Definition 3.3, $w_{t+1}^{\text{Grams}}$ is updated from $w_t$ using Definition 5.1. Then we have the following:*

- *Part 1. It holds that*

$$\Delta \mathcal{L}_{w_{t+1}^{\text{Grams}}, w_t} \leq -\eta_t \langle |g_t|, |u_t| \rangle + \frac{L \eta_t^2}{2} \|u_t\|_2^2. \tag{3}$$

- *Part 2. It holds that $\Delta \mathcal{L}_{w_{t+1}^{\text{Grams}}, w_t} \geq -\eta_t \langle |g_t|, |u_t| \rangle$.*

- *Part 3. If $\eta_t \leq \frac{2}{L \|u_t\|^2} \langle |g_t|, |u_t| \rangle$, then we have $\Delta \mathcal{L}_{w_{t+1}^{\text{Grams}}, w_t} \leq 0$.*

Then, we compare the loss descent between Grams and C-Adam.

**Theorem 5.4** (Loss Descent Comparison, informal version of Theorem D.3). *Suppose that $\mathcal{L} : \mathbb{R}^d \to \mathbb{R}$ is $L$-smooth. For any parameter vector $w$ at optimization step $t$, let $w_t^{\text{Grams}}$ and $w_t^{\text{C}}$ be the update of Grams in Definition 5.1 and Cautious optimizers in Definition 3.2, respectively. If the stepsize $\eta_t$ satisfies $\eta_t \leq \frac{2}{L \|u_t\|^2} \cdot \min\{\langle u_t \circ g_t, \mathbf{1}_{u_t \circ g_t \geq 0} \rangle, \langle u_t \circ g_t, \mathbf{1}_{u_t \circ g_t < 0} \rangle\}$, then we have $\Delta \mathcal{L}_{w_{t+1}^{\text{Grams}}, w_t} \leq \Delta \mathcal{L}_{w_{t+1}^{\text{C}}, w_t} \leq 0$.*

**Remark 5.5.** *Theorem 5.4 shows that Grams achieves strictly better descent in the loss landscape in the discrete analysis compared to Cautious optimizers. This theoretical guarantee suggests that Grams may converge faster and achieve better minima in practice.*

### 5.3 HAMILTONIAN DYNAMICS

In this subsection, we present the Grams Hamiltonian dynamics, which builds upon the augmented Hamiltonian framework to analyze optimization algorithms. By leveraging this framework, we show that the Grams optimizer achieves a monotonic descent of the Hamiltonian and the loss function, with a descent speed that is provably equal to or faster than C-Adam. This highlights Grams' efficiency and robustness in dynamic optimization landscapes. The formal definition is provided below.

**Definition 5.6** (Grams Hamiltonian Dynamics). *We could modify Hamiltonian dynamics with Grams' optimizing scheme,*

$$\frac{\mathrm{d}}{\mathrm{d}t}w_t := -\operatorname{sign}(\nabla\mathcal{L}(w_t) \circ |\nabla\mathcal{K}(s_t)|) - \Phi_t(\nabla\mathcal{L}(w_t)),$$

$$\frac{\mathrm{d}}{\mathrm{d}t}s_t := \nabla\mathcal{L}(w_t) - \Psi_t(\nabla\mathcal{K}(s_t)),$$

*where $|\cdot|$ denotes element-wise absolute value, $\circ$ is the Hadamard product, and $\Phi_t, \Psi_t$ are scaling functions.*

The convergence properties of Grams within the Hamiltonian dynamics framework are formalized in the theorem below.

**Theorem 5.7** (Convergence of Grams Hamiltonian Dynamics, informal version of Theorem E.3). *Following the dynamics in Definition 5.6, we have*

$$\Delta_H^{Grams}(w_t, s_t) := \frac{\mathrm{d}}{\mathrm{d}t}H(w_t, s_t) \le 0,$$

$$\Delta_{\mathcal{L}}^{Grams}(w_t) := \frac{\mathrm{d}}{\mathrm{d}t}\mathcal{L}(w_t) \le -\Delta_{\mathcal{L}}(w_t, s_t),$$

*where $\Delta_{H_t}(w_t, s_t)$ and $\Delta_{\mathcal{L}_t}(w_t, s_t)$ represent the decreasing rates of $H$ and $\mathcal{L}$ in accordance with the system in Definition E.1.*

Based on this theorem, we compare the convergence rates of Grams and Cautious optimizers in the context of Hamiltonian dynamics. The following theorem demonstrates that Grams achieves a faster or equal rate of loss descent compared to Cautious optimizers, highlighting its efficiency in optimization.

**Theorem 5.8** (Convergence Comparison of Hamiltonian Dynamics between Grams and Cautious Optimizers, informal version of Theorem E.4). *From Theorem 5.7 and E.2, recall $\Delta_{\mathcal{L}}^{Grams}(w_t)$ and $\Delta_{\mathcal{L}}^{C}(w_t)$, we have $\Delta_{\mathcal{L}}^{Grams}(w_t) \le \Delta_{\mathcal{L}}^{C}(w_t)$.*

**Remark 5.9.** *Theorem 5.8 illustrates the faster loss decreasing speed in the Grams Hamiltonian dynamic system, compared to Cautious's counterpart.*

Building on this comparison, we now state a corollary from (Liang et al., 2024) that establishes the convergence of bounded solutions in Hamiltonian systems to stationary points of the augmented loss.

**Corollary 5.10** (Corollary 2.4 in (Liang et al., 2024)). *Assume that $\langle x, \Psi(x)\rangle$ is positive definite for all $x \in \mathbb{R}^d$, $\Psi(0) = 0$, and that $H(w, s) = \mathcal{L}(w) + \mathcal{K}(s)$ is differentiable. Then, the bounded solutions of the original system Eq. (15) converge to a stationary point of $H(w, s)$. Similarly, the bounded solutions of Definition 5.6 also converge to a stationary point of $H(w, s)$.*

## 6 EXPERIMENTS

We conducted comprehensive experiments across both pre-training and fine-tuning stages to evaluate the performance of our proposed Grams optimizer. Comparisons were made against several baseline optimizers, including Adam (Kingma & Ba, 2014), Lion (Chen et al., 2024), C-Adam, C-Lion (Liang et al., 2024), and, in some experiments, RMSprop (Hinton et al., 2012; Ruder, 2016).

The details and hyperparameters of our experiments can be found in Section F.

## 6.1 HYPERPARAMETERS

We conduct a hyperparameter grid search to evaluate Grams' sensitivity to $\beta_1$ and $\beta_2$ by training a VAE (Kingma & Welling, 2013) on the CIFAR-10 (Krizhevsky, 2009) dataset. Each model is trained for 10 epochs, and the mean squared error (MSE) loss is reported as the evaluation metric. See Figure 2 for the results.

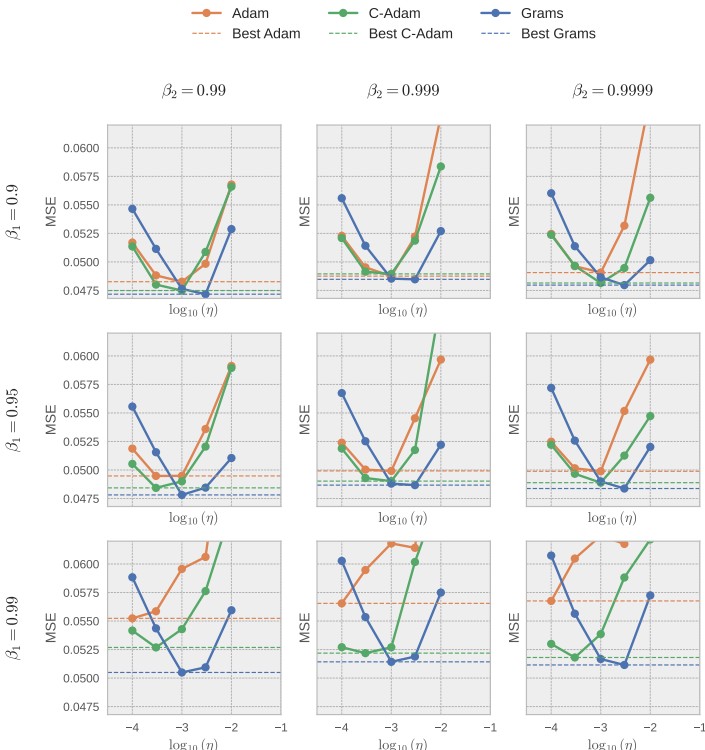

Figure 2: Evaluation loss for VAE training experiments with grid search.

From Figure 2, we observe that for each hyperparameter configuration, Grams achieves the lowest MSE loss, indicating superior reconstruction quality. It also attains the lowest overall MSE across all setups when $\beta_1 = 0.9$, $\beta_2 = 0.99$, and $\eta = 3 \times 10^{-3}$. These results demonstrate that Grams consistently outperforms both Adam and C-Adam across all tested $\beta$ configurations.

## 6.2 PRE-TRAINING

We train the Llama 60M model (Dubey et al., 2024) using the first $2,048,000$ rows of data from English subset of the C4 dataset (Raffel et al., 2020) to assess Grams' optimization capability for Transformer-based (Vaswani et al., 2017) natural language generation (NLG) tasks. Moreover, we trained and evaluated the WideResNet-50-2 model (Zagoruyko & Komodakis, 2016) on the CIFAR-10 dataset (Krizhevsky, 2009). See Table 1 for experiment results.

Table 1 reports the validation results of pre-training experiments, where the Grams optimizer achieves the lowest perplexity on the Llama language model and the highest accuracy on WideResNet. These results highlight the effectiveness of Grams in both language and vision domains. While C-Adam and C-Lion show improvements over their respective baselines, Adam and Lion, Grams consistently outperforms all variants, indicating superior convergence and generalization. Notably, as the AMS parameter $\lambda$ decreases from 1 to $-1$, WRN accuracy steadily improves, suggesting that smaller $\lambda$ values enhance adaptation in vision tasks.

Table 1: Validation results of Llama and WideResNet (WRN) pre-training. Llama results are reported in perplexity (lower is better), and WRN results in accuracy (higher is better). For all AMS, $\beta_3 = 0$.

| Optimizer | Llama PPL↓ | WRN Acc↑ |
|---|---|---|
| RMSprop | N/A | 84.47% |
| AMS $\lambda = 1$ (Adam) | 49.83 | 87.56% |
| AMS $\lambda = 0.5$ | N/A | 88.10% |
| AMS $\lambda = 0$ (C-Adam) | 43.21 | 88.78% |
| AMS $\lambda = -0.5$ | N/A | 89.34% |
| AMS $\lambda = -1$ (Grams) | **38.60** | **90.55%** |
| Lion | 50.25 | 89.21% |
| C-Lion | 53.21 | 89.42% |

## 6.3 FINE-TUNING

We performed full fine-tuning (FT) experiments on the Llama 3.2 1B model (Dubey et al., 2024), and SORSA method (Cao, 2024) for parameter efficient fine-tuning (PEFT) experiments on Llama 3.2 3B model, both using the first 100,000 rows of data from the MetaMathQA dataset (Yu et al., 2023). To evaluate the models, we measured accuracy on the GSM-8K dataset (Cobbe et al., 2021) and MATH dataset (Hendrycks et al., 2021), respectively. The results are reported in Table 2.

Table 2: Evaluation results for Llama fine-tuning experiments. Results are reported using Pass@1 accuracy, where higher values indicate better performance.

| Optimizer | GSM-8K | MATH |
|---|---|---|
| AMS $\lambda = 1$ (Adam) | 48.90% | **17.80%** |
| AMS $\lambda = 0$ (C-Adam) | 49.81% | 16.62% |
| AMS $\lambda = -1$ (Grams) | **51.02%** | **17.80%** |

The results in Table 2 showcase the performance of different optimizers during the fine-tuning experiments on the Llama models using the MetaMathQA dataset. Among the optimizers, Grams achieved the highest or tied-highest accuracy on both experiments. These results highlight the effectiveness of Grams in fine-tuning tasks, particularly in improving the model's ability to handle complex datasets like GSM-8K. The superior performance of Grams demonstrates its capacity to achieve better generalization and optimization efficiency in fine-tuning scenarios.

## 7 CONCLUSION AND FUTURE WORK

In this paper, we introduced Gradient Descent with Adaptive Momentum Scaling (Grams), a novel optimization algorithm designed to decouple the direction and magnitude of parameter updates. By leveraging this decoupling, Grams demonstrated superior performance in both theoretical convergence guarantees and empirical evaluations, outperforming state-of-the-art optimizers such as Adam (Loshchilov & Hutter, 2017), Lion (Chen et al., 2024), and their Cautious variants (Liang et al., 2024). The results across various tasks highlight Grams' potential as a transformative approach for efficiently training large language models.

Building on the promising results of Grams, future work will focus on integrating ideas from recent advancements such as ADOPT (Taniguchi et al., 2024), Schedule Free (Defazio et al., 2024), and SOAP-Muon (Vyas et al., 2025) methods. Incorporating the ADOPT and schedule-free learning rate adjustment strategies might improve Grams' robustness and performance across diverse tasks and architectures. By blending these complementary innovations with the core principles of Grams, we aim to develop an even more versatile and efficient optimization framework for large language model training and fine-tuning.

ETHIC STATEMENT

This paper does not involve human subjects, personally identifiable data, or sensitive applications. We do not foresee direct ethical risks. We follow the ICLR Code of Ethics and affirm that all aspects of this research comply with the principles of fairness, transparency, and integrity.

REPRODUCIBILITY STATEMENT

We ensure reproducibility on both theoretical and empirical fronts. For theory, we include all formal assumptions, definitions, and complete proofs in the appendix. For experiments, we describe model architectures, datasets, preprocessing steps, hyperparameters, and training details in the main text and appendix. Code and scripts are provided in the supplementary materials to replicate the empirical results.

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

# Appendix

**Roadmap.** In the appendix, we first provide more context on optimization in Section A. Then, we show the formal version of equivalence of AMS frameworks in Section B. We then provide some useful facts in Section C, which are utilized in the results. Section D presents a formal analysis of loss descent for Grams optimizers. In Section E, we illustrate the the property of Grams optimizer in the landscape of Hamiltonian dynamics. Finally, we list the details of our experiments in Section F.

## A PRELIMINARY

In this section, we provide more context on optimization.

### A.1 BACKGROUNDS ON OPTIMIZATION

We define the $L$-smoothness of functions as below.

**Definition A.1** ($L$-smooth). *We say that a function $f : \mathbb{R}^d \to \mathbb{R}$ is $L$-smooth if $\|\nabla f(x_1) - \nabla f(x_2)\|_2 \le L\|x_1 - x_2\|_2$ for all $x_1, x_2 \in \mathbb{R}^d$.*

We state a common fact of $L$-smooth functions as follow.

**Fact A.2.** *If a function $f : \mathbb{R}^d \to \mathbb{R}$ is $L$-smooth, then we have*

$$f(x_2) \le f(x_1) + \langle \nabla f(x_1), x_2 - x_1 \rangle + \frac{L}{2}\|x_2 - x_1\|_2^2,$$

$$f(x_2) \ge f(x_1) + \langle \nabla f(x_1), x_2 - x_1 \rangle - \frac{L}{2}\|x_2 - x_1\|_2^2.$$

We also define PL-condition as below.

**Definition A.3** (PL-condition). *A function $f : \mathbb{R}^d \to \mathbb{R}$ satisfies the $\mu$-Polyak–Łojasiewicz (PL) condition with constant $\mu > 0$ if the following inequality holds for all $x \in \mathbb{R}^d$:*

$$\|\nabla f(x)\|^2 \ge 2\mu(f(x) - f^*),$$

*where $f^*$ is the minimum value of the function $f$, i.e., $f^* = \inf_{x \in \mathbb{R}^d} f(x)$.*

### A.2 ADAM OPTIMIZER

Adam (Adaptive Moment Estimation) (Kingma & Ba, 2014) is a widely-used optimizer that combines the benefits of RMSprop (Hinton et al., 2012) and momentum by maintaining both first and second moment estimates of the gradients. The algorithm adapts the learning rates for each parameter using these estimates. See Algorithm 2 for the pseudo-code of Adam.

---

**Algorithm 2** Adam (Kingma & Ba, 2014)

---

**Require:** parameter $w$, step sizes $\{\eta_t\}$, dampening factors $\beta_1, \beta_2 \in [0, 1)$, $\epsilon > 0$, weight decay $\gamma \ge 0$
1: Initialize $t = 0$, $m_0 = v_0 = \mathbf{0}$
2: **while** $w_t$ not converged **do**
3:      $t \leftarrow t + 1$
4:      $g_t \leftarrow \nabla_w \mathcal{L}_t(w_{t-1})$
5:      $m_t \leftarrow \beta_1 m_{t-1} + (1 - \beta_1)g_t$
6:      $v_t \leftarrow \beta_2 v_{t-1} + (1 - \beta_2)g_t^2$
7:      $\widehat{m}_t \leftarrow m_t/(1 - \beta_1^t)$
8:      $\widehat{v}_t \leftarrow v_t/(1 - \beta_2^t)$
9:      $u_t \leftarrow \widehat{m}_t/(\sqrt{\widehat{v}_t} + \epsilon)$
10:      $w_t \leftarrow w_{t-1} - \epsilon_t u_t$
11:      $w_t \leftarrow w_t - \epsilon_t \gamma w_t$                ▷ Add weight decay (Loshchilov & Hutter, 2017)
12: **end while**

---

**Definition A.4** (Adam). *The parameter update rule for Adam is given by:*

$$m_t := \beta_1 m_{t-1} + (1 - \beta_1) g_t$$
$$v_t := \beta_2 v_{t-1} + (1 - \beta_2) g_t^2$$
$$\widehat{m}_t := \frac{m_t}{1 - \beta_1^t}$$
$$\widehat{v}_t := \frac{v_t}{1 - \beta_2^t}$$
$$u_t := \frac{\widehat{m}_t}{\sqrt{\widehat{v}_t} + \epsilon}$$
$$w_{t+1} := w_t - \eta_t u_t,$$

*where $w_t$ is the weight at time step $t$, $m_t$ and $v_t$ are the first and second momentum estimates respectively, $g_t = \nabla_w \mathcal{L}_t(w_{t-1})$ is the current gradient, $\beta_1$ and $\beta_2$ are decay rates for the moment estimates, $\epsilon$ is a small constant for numerical stability, and $\eta_t$ is the learning rate at step $t$.*

### A.3 LION OPTIMIZER

Evolved Sign Momentum (Lion) (Chen et al., 2024) is an efficient optimizer that leverages momentum and sign-based updates. Lion's key innovation lies in its update rule, which combines both current and momentum gradients through sign operations.

**Definition A.5** (Lion Parameter Update). *The parameter update rule for Lion is given by:*

$$u_t := \text{sign}(\beta_1 m_{t-1} + (1 - \beta_1) g_t)$$
$$w_t := w_{t-1} - \eta_t \cdot u_t$$
$$m_t := \beta_2 m_{t-1} + (1 - \beta_2) g_t,$$

*where $w_t$ is the weight at time step $t$, $m_{t-1}$ is the momentum term, $g_t = \nabla_w \mathcal{L}_t(w_{t-1})$ is the current gradient, $\beta_1$ and $\beta_2$ are the momentum coefficients, $\eta_t$ is the learning rate at step $t$, and $\text{sign}$ is defined in Definition 3.1,*

Lion's efficiency stems from its memory-efficient design - it only needs to maintain a single momentum term and operates primarily through sign operations. This makes it particularly suitable for large-scale training where memory constraints are significant. The optimizer has demonstrated strong performance in training large language models and vision transformers, often achieving comparable or better results than Adam while using less memory.

## B EQUIVALENCE OF ADAPTIVE MOMENTUM SCALING

**Corollary B.1** (Formal version of Corollary 4.1). *By choosing $\beta_3 = \beta_1$, AMS is equivalent with Adam (Definition A.4).*

*Proof.* **Case 1.** $\beta_3 = \beta_1$,

$$\begin{aligned} s_t &= \beta_3 s_{t-1} + (1 - \beta_3) g_t \\ &= \beta_1 m_{t-1} + (1 - \beta_1) g_t \\ &= m_t, \end{aligned} \tag{4}$$

Then, by the definition of $\phi_t$,

$$\begin{aligned} \phi_t &= \mathbf{1}_{u_t \circ s_t \geq 0} \\ &= \mathbf{1}_{u_t \circ m_t \geq 0} \\ &= \mathbf{1}_{\frac{m_t}{(1 - \beta_1^t)(\sqrt{\widehat{v}} + \epsilon)} \circ m_t \geq 0} \\ &= \mathbf{1}_d, \end{aligned}$$

where the first step follows the definition of $\phi$, the second step follows from Eq. (4), the third step follows the definition of $u_t$, and the last step holds because $(1 - \beta_1^t)(\sqrt{\widehat{v}} + \epsilon)_i \geq 0 \quad \forall i \in d$.

Likewise, we show,

$$\psi_t = \lambda \cdot \mathbf{1}_{u_t \circ s_t < 0}$$
$$= \mathbf{1}_{\frac{m_t}{(1-\beta_1^t)(\sqrt{\hat{v}}+\epsilon)} \circ m_t < 0}$$
$$= \mathbf{0}_d,$$

where the first step follows the definition of $\psi$, second step follows from Eq. (4), the last step holds because $(1 - \beta_1^t)(\sqrt{\hat{v}} + \epsilon)_i \geq 0 \quad \forall i \in d$.

By the definition of $\hat{u}_t$,

$$\hat{u}_t = (\phi_t + \psi_t) \circ u_t$$
$$= (\mathbf{1}_d + \mathbf{0}_d) \circ u_t$$
$$= u_t,$$

where the second step substitute $\phi_t$ and $\psi_t$.

By the definition of $\hat{\eta}_t$,

$$\hat{\eta} = \eta \frac{d}{\|\phi_t\|_1 + \|\|\psi_t\|_1\|}$$
$$= \eta \frac{d}{\|\mathbf{1}_d\|_1 + \|\|\mathbf{0}_d\|_1\|}$$
$$= \eta \frac{d}{d + 0}$$
$$= \eta,$$

where the second step substitute $\phi$ and $\psi$, the third step follows the definition of $\ell_1$ norm.

**Case 2.** $\lambda = 1$.

By the definition of $\hat{u}_t$,

$$\hat{u}_t = (\phi_t + \psi_t) \circ u_t$$
$$= (\mathbf{1}_{u_t \circ s_t \geq 0} + \mathbf{1}_{u_t \circ s_t < 0}) \circ u_t$$
$$= u_t,$$

where the second step follows from the definition of $\phi_t$ and $\psi_t$, the third step holds because the fact that $\mathbf{1}_{u_t \circ s_t \geq 0} + \mathbf{1}_{u_t \circ s_t < 0} = \mathbf{1}_d$.

By the definition of $\hat{\eta}_t$,

$$\hat{\eta} = \eta \frac{d}{\|\phi_t\|_1 + \|\|\psi_t\|_1\|}$$
$$= \eta \frac{d}{\|\mathbf{1}_{u_t \circ s_t \geq 0}\|_1 + \|\|\mathbf{1}_{u_t \circ s_t < 0}\|_1\|}$$
$$= \eta \frac{d}{d}$$
$$= \eta.$$

Thus we complete the proof. $\square$

**Corollary B.2** (Formal version of Corollary 4.2). *By choosing $\beta_3 = 0$ and $\lambda = 0$, AMS is equivalent with Cautious Adam (Definition 3.2).*

*Proof.* When $\beta_3 = 0$,

$$s_t = \beta_3 s_{t-1} + (1 - \beta_3)g_t$$
$$= g_t. \tag{5}$$

Then, by the definition of $\phi_t$,

$$\phi_t = \mathbf{1}_{u_t \circ s_t \geq 0}$$
$$= \mathbf{1}_{u_t \circ g_t \geq 0}$$

where the second step follows Eq. (5).

Likewise, since $\lambda = 0$, by the definition of $\psi_t$, we show

$$\psi_t = \lambda \cdot \mathbf{1}_{u_t \circ s_t < 0}$$
$$= 0 \cdot \mathbf{1}_{u_t \circ g_t < 0}$$
$$= \mathbf{0}_d.$$

By the definition of $\widehat{u}_t$,

$$\widehat{u}_t = (\phi_t + \psi_t) \circ u_t$$
$$= \mathbf{1}_{u_t \circ g_t \geq 0} \circ u_t,$$

where the second step substitutes $\phi_t$ and $\psi_t$.

By the definition of $\widehat{\eta}$,

$$\widehat{\eta} = \eta \frac{d}{\|\phi_t\|_1 + \|\|\psi_t\|_1\|}$$
$$= \eta \frac{d}{\|\phi_t\|_1}$$

where the second step follows $\psi_t = \mathbf{0}_d$ and the definition of $\ell_1$ norm.

Thus we complete the proof. □

**Corollary B.3** (Formal version of Corollary 5.2). *By choosing $\beta_3 = 0$ and $\lambda = -1$, the update rule of AMS is equivalent with:*

$$m_t := \beta_1 m_{t-1} + (1 - \beta_1)g_t,$$
$$v_t := \beta_2 v_{t-1} + (1 - \beta_2)g_t^2,$$
$$\widehat{m}_t := \frac{m_t}{1 - \beta_1^t},$$
$$\widehat{v}_t := \frac{v_t}{1 - \beta_2^t},$$
$$u_t := \frac{\widehat{m}_t}{\sqrt{\widehat{v}_t} + \epsilon},$$
$$\widehat{u}_t := \text{sign}(g_t) \circ |u_t|,$$
$$w_t := w_{t-1} - \eta_t \widehat{u}_t, \tag{6}$$

*Proof.* When $\beta_3 = 0$, Eq. (5) holds.

Then, by the definition of $\phi_t$,

$$\phi_t = \mathbf{1}_{u_t \circ s_t \geq 0}$$
$$= \mathbf{1}_{u_t \circ g_t \geq 0},$$

where the second step follows from Eq. (5).

Since $\lambda = -1$, by definition of $\psi_t$,

$$\psi_t = \lambda \cdot \mathbf{1}_{u_t \circ s_t < 0}$$
$$= -\mathbf{1}_{u_t \circ g_t < 0},$$

where the second step follows from Eq. (5).

Next, by the definition of $\widehat{u}_t$,

$$\widehat{u}_t = (\phi_t + \psi_t) \circ u_t$$

$$= (\mathbf{1}_{u_t \circ g_t \geq 0} - \mathbf{1}_{u_t \circ g_t < 0}) \circ u_t,$$
$$= \mathbf{1}_{u_t \circ g_t \geq 0} \circ u_t + \mathbf{1}_{u_t \circ g_t < 0} \circ (-u_t)$$
$$= \text{sign}(u_t \circ g_t) \circ u_t$$
$$= \text{sign}(g_t) \circ |u_t|.$$

where the second step substitutes $\phi_t$ and $\psi_t$.

By the definition of $\widehat{\eta}$,

$$\widehat{\eta} = \eta \frac{d}{\|\phi_t\|_1 + \|\|\psi_t\|_1\|}$$
$$= \eta \frac{d}{\|\phi_t\|_1 - \|\psi_t\|_1}$$
$$= \eta.$$

where the second step follows the fact that $\lambda = -1$, and the last step from definition of $\ell_1$-norm.

Thus we complete the proof. $\qquad\square$

## C  USEFUL FACTS

**Fact C.1.** *Given vectors $a, b, c \in \mathbb{R}^d$, we have*

$$\langle a, b \circ c \rangle = \langle a \circ b, c \rangle.$$

**Fact C.2.** *Let two vectors $a, b \in \mathbb{R}^n$, then:*

$$\langle a, -\text{sign}(a) \circ |b| \rangle = -\langle |a|, |b| \rangle$$

*Proof.* For the left side of the equation:

$$\langle a, -\text{sign}(a) \circ |b| \rangle = \sum_{i=1}^{n} -a_i \text{sign}(a_i)|b|_i$$
$$= -\sum_{i=1}^{n} |a|_i|b|_i$$
$$= -\langle |a|, |b| \rangle$$

where the first step comes from the definition of inner product, the second step uses Fact C.5, and the final step uses the definition of inner product again. $\qquad\square$

**Fact C.3.** *Let two vectors $a, b \in \mathbb{R}^n$, then:*

$$\langle a, b \rangle - \langle |a|, |b| \rangle \leq 0.$$

*Proof.*

$$\langle a, b \rangle - \langle |a|, |b| \rangle = \sum_{i=1}^{n} a_i b_i - |a|_i|b|_i$$
$$= \sum_{i=1}^{n} \begin{cases} 0 & \text{if } a_i \text{ and } b_i \text{ have the same sign} \\ -2|a_i||b_i| & \text{if } a_i \text{ and } b_i \text{ have opposite signs} \end{cases}$$
$$\leq 0,$$

where the first step uses the definition of inner product, the second step discusses the only two cases we have for signs, and the final inequality comes from basic algebra. $\qquad\square$

**Fact C.4.** *Let $x = a \circ b$ be an element-wise product of two vectors $a, b \in \mathbb{R}^n$, then:*

$$\langle a, b \rangle - \langle |a|, |b| \rangle - \langle a \circ b, \mathbf{1} - \mathbf{1}_{a \circ b > 0} \rangle \leq 0$$

*Proof.*

$$\langle a, b \rangle - \langle |a|, |b| \rangle - \langle a \circ b, \mathbf{1} - \mathbf{1}_{a \circ b > 0} \rangle$$

$$= \sum_{i=1}^{n} a_i b_i - \sum_{i=1}^{n} |a_i||b_i| - \left( \sum_{i=1}^{n} a_i b_i - \sum_{i:a_i b_i > 0} a_i b_i \right)$$

$$= \sum_{i:a_i b_i > 0} a_i b_i - \sum_{i=1}^{n} |a_i||b_i|,$$

where the first step expands the terms, and the second step simplifies by splitting the sum based on the sign of $a_i b_i$.

If all $a_i b_i \geq 0$, then $\sum_{i:a_i b_i > 0}^{n} a_i b_i = \sum_{i=1}^{n} |a_i||b_i|$, so the expression is 0. Otherwise, $\sum_{i=1}^{n} |a_i||b_i| > \sum_{i:a_i b_i > 0}^{n} a_i b_i$, so the expression is negative.

Thus,

$$\langle a, b \rangle - \langle |a|, |b| \rangle - \langle a \circ b, \mathbf{1}_d - \mathbf{1}_{a \circ b > 0} \rangle = \sum_{i:a_i b_i > 0} a_i b_i - \sum_{i=1}^{n} |a_i||b_i| \leq 0.$$

The proof is complete. □

**Fact C.5.** *Given a scalar $a \in \mathbb{R}$, we have:*

$$a \cdot \mathrm{sign}(a) = |a|.$$

*Proof.* Let $a \in \mathbb{R}$. By Definition 3.1:

$$\mathrm{sign}(a) = \begin{cases} 1, & \text{if } a > 0, \\ 0, & \text{if } a = 0, \\ -1, & \text{if } a < 0. \end{cases}$$

Consider the following cases:

- If $a > 0$, then $\mathrm{sign}(a) = 1$, so:
$$a \cdot \mathrm{sign}(a) = a \cdot 1 = a = |a|.$$

- If $a = 0$, then $\mathrm{sign}(a) = 0$, so:
$$a \cdot \mathrm{sign}(a) = 0 \cdot 0 = 0 = |a|.$$

- If $a < 0$, then $\mathrm{sign}(a) = -1$, so:
$$a \cdot \mathrm{sign}(a) = a \cdot (-1) = -a = |a|.$$

Thus, in all cases, $a \cdot \mathrm{sign}(a) = |a|$. □

**Fact C.6.** *Given a vector $a = (a_1, a_2, \ldots, a_n) \in \mathbb{R}^n$, we have:*

$$a \circ \mathrm{sign}(a) = |a|,$$

*where the operations are applied component-wise.*

*Proof.* Let $a = (a_1, a_2, \ldots, a_n) \in \mathbb{R}^n$. By Definition 3.1, the sign function is applied component-wise:

$$\mathrm{sign}(a) = (\mathrm{sign}(a_1), \mathrm{sign}(a_2), \ldots, \mathrm{sign}(a_n)).$$

Expanding the Hadamard product $a \circ \mathrm{sign}(a)$ component-wise:

$$a \circ \mathrm{sign}(a) = (a_1 \cdot \mathrm{sign}(a_1), a_2 \cdot \mathrm{sign}(a_2), \ldots, a_n \cdot \mathrm{sign}(a_n)).$$

By Fact C.5 (the scalar version), for each $i$:

$$a_i \cdot \mathrm{sign}(a_i) = |a_i|.$$

Thus:

$$a \circ \mathrm{sign}(a) = (|a_1|, |a_2|, \ldots, |a_n|) = |a|,$$

where the absolute value $|a|$ is applied component-wise. □

# D  Loss Descent

**Lemma D.1** (Formal version of Lemma 3.4). *Suppose that $\mathcal{L} : \mathbb{R}^d \to \mathbb{R}$ is $L$-smooth. Let $\Delta\mathcal{L}_{w_{t+1}^{\mathrm{C}}, w_t}$ be defined in Definition 3.3, $w_{t+1}^{\mathrm{C}}$ is updated from $w_t$ using Definition 3.2. Then we have the followings:*

- *Part 1. It holds that*

$$\Delta\mathcal{L}_{w_{t+1}^{\mathrm{C}}, w_t} \leq -\eta_t \langle u_t \circ g_t, \mathbf{1}_{u_t \circ g_t \geq 0}\rangle + \frac{L\eta_t^2}{2}\|u_t\|_2^2, \tag{7}$$

- *Part 2. It holds that*

$$\Delta\mathcal{L}_{w_{t+1}^{\mathrm{C}}, w_t} \geq -\eta_t \langle u_t \circ g_t, \mathbf{1}_{u_t \circ g_t \geq 0}\rangle.$$

- *Part 3. If $\eta_t \leq \frac{2}{L\|u_t\|_2^2}\langle u_t \circ g_t, \mathbf{1}_{u_t \circ g_t \geq 0}\rangle$, then $\Delta\mathcal{L}_{w_{t+1}^{\mathrm{C}}, w_t} \leq 0$.*

*Proof.* **Proof of Part 1.** We can show that

$$
\begin{aligned}
\Delta\mathcal{L}_{w_{t+1}^{\mathrm{C}}, w_t} &= \mathcal{L}(w_{t+1}) - \mathcal{L}(w_t) \\
&\leq \mathcal{L}(w_t) + \langle g_t, w_{t+1} - w_t\rangle + \frac{L}{2}\|w_{t+1} - w_t\|_2^2 - \mathcal{L}(w_t) \\
&= \langle g_t, w_{t+1} - w_t\rangle + \frac{L}{2}\|w_{t+1} - w_t\|_2^2 \\
&= \langle g_t, -\eta_t u_t \circ \mathbf{1}_{u_t \circ g_t \geq 0}\rangle + \frac{L}{2}\|\eta_t u_t \circ \mathbf{1}_{u_t \circ g_t \geq 0}\|_2^2 \\
&= -\eta_t \langle u_t \circ g_t, \mathbf{1}_{u_t \circ g_t \geq 0}\rangle + \frac{L}{2}\|\eta_t u_t \circ \mathbf{1}_{u_t \circ g_t \geq 0}\|_2^2 \\
&\leq -\eta_t \langle u_t \circ g_t, \mathbf{1}_{u_t \circ g_t \geq 0}\rangle + \frac{L\eta_t^2}{2}\|u_t\|_2^2
\end{aligned}
\tag{8}
$$

where the first step follows from Definition 3.3, the second step follows from that $\mathcal{L}$ is $L$-smooth and Fact A.2, the third step follows from basic algebra, the fourth step follows from Definition 3.2, the fifth step follows from Fact C.1, and the last step follows from basic algebra.

**Proof of Part 2.** Next, we can show that

$$
\begin{aligned}
\Delta\mathcal{L}_{w_{t+1}^{\mathrm{C}}, w_t} &= \mathcal{L}(w_{t+1}) - \mathcal{L}(w_t) \\
&\geq \mathcal{L}(w_t) + \langle g_t, w_{t+1} + w_t\rangle - \frac{L}{2}\|w_{t+1} - w_t\|_2^2 - \mathcal{L}(w_t) \\
&\geq \langle g_t, w_{t+1} - w_t\rangle \\
&= \langle g_t, -\eta_t u_t \circ \mathbf{1}_{u_t \circ g_t \geq 0}\rangle \\
&= -\eta_t \langle u_t \circ g_t, \mathbf{1}_{u_t \circ g_t \geq 0}\rangle
\end{aligned}
\tag{9}
$$

where the first step follows from Definition 3.3, the second step follows from that $\mathcal{L}$ is $L$-smooth and Fact A.2, the third step follows from basic algebra, the fourth step follows from Definition 3.2, the last step follows from Fact C.1.

**Proof of Part 3.** By rearranging the Eq. (8), it is clear that if $\eta_t \leq \frac{2}{L\|u_t\|_2^2}\langle u_t \circ g_t, \mathbf{1}_{u_t \circ g_t \geq 0}\rangle$, then we have $\Delta\mathcal{L}_{w_{t+1}^{\mathrm{C}}, w_t} \leq 0$. $\square$

**Lemma D.2** (Formal version of Lemma 5.3). *Suppose that $\mathcal{L} : \mathbb{R}^d \to \mathbb{R}$ is $L$-smooth. Let $\Delta\mathcal{L}_{w_{t+1}^{\mathrm{Grams}}, w_t}$ be defined in Definition 3.3, $w_{t+1}^{\mathrm{Grams}}$ is updated from $w_t$ using Definition 5.1. Then we have the following:*

- *Part 1. It holds that*

$$\Delta\mathcal{L}_{w_{t+1}^{\mathrm{Grams}}, w_t} \leq -\eta_t \langle |g_t|, |u_t|\rangle + \frac{L\eta_t^2}{2}\|u_t\|_2^2. \tag{10}$$

- *Part 2. It holds that*

$$\Delta\mathcal{L}_{w_{t+1}^{\mathrm{Grams}}, w_t} \geq -\eta_t \langle |g_t|, |u_t| \rangle.$$

- *Part 3. If $\eta_t \leq \frac{2}{L\|u_t\|^2} \langle |g_t|, |u_t| \rangle$, then we have $\Delta\mathcal{L}_{w_{t+1}^{\mathrm{Grams}}, w_t} \leq 0$.*

*Proof.* **Proof of Part 1.** We can show that

$$\Delta\mathcal{L}_{w_{t+1}^{\mathrm{Grams}}, w_t} = \mathcal{L}(w_{t+1}) - \mathcal{L}(w_t)$$

$$\leq \mathcal{L}(w_t) + \langle g_t, w_{t+1} - w_t \rangle + \frac{L}{2}\|w_{t+1} - w_t\|_2^2 - \mathcal{L}(w_t)$$

$$= \langle g_t, w_{t+1} - w_t \rangle + \frac{L}{2}\|w_{t+1} - w_t\|_2^2$$

$$= \langle g_t, -\eta_t \cdot \mathrm{sign}(g_t) \circ |u_t| \rangle + \frac{L}{2}\|\eta_t \cdot \mathrm{sign}(g_t) \circ |u_t|\|_2^2$$

$$= -\eta_t \langle g_t \circ \mathrm{sign}(g_t), |u_t| \rangle + \frac{L}{2}\|\eta_t u_t\|_2^2$$

$$\leq -\eta_t \langle |g_t|, |u_t| \rangle + \frac{L\eta_t^2}{2}\|u_t\|_2^2 \tag{11}$$

where the first step follows from Definition 3.3, the second step follows from that $\mathcal{L}$ is $L$-smooth and Fact A.2, the third step follows from basic algebra, the fourth step follows from Definition 5.1, the fifth step follows from the Fact C.1, and the last step follows from $g_t \circ \mathrm{sign}(g_t) = |g_t|$.

**Proof of Part 2.** Next, we can show that

$$\Delta\mathcal{L}_{w_{t+1}^{\mathrm{Grams}}, w_t} = \mathcal{L}(w_{t+1}) - \mathcal{L}(w_t)$$

$$\geq \mathcal{L}(w_t) + \langle g_t, w_{t+1} + w_t \rangle - \frac{L}{2}\|w_{t+1} - w_t\|_2^2 - \mathcal{L}(w_t)$$

$$\geq \langle g_t, w_{t+1} - w_t \rangle$$

$$= \langle g_t, -\eta_t \cdot \mathrm{sign}(g_t) \circ |u_t| \rangle$$

$$= -\eta_t \langle |g_t|, |u_t| \rangle \tag{12}$$

where the first step follows from Definition 3.3, the second step follows from that $\mathcal{L}$ is $L$-smooth and Fact A.2, the third step follows from basic algebra, the fourth step follows from Definition 5.1, the last step follows from the Fact C.1 and Fact C.6.

**Proof of Part 3.** By rearranging the Eq. (11), it is clear that if $\eta_t \leq \frac{2}{L\|u_t\|_2^2} \langle |g_T|, |u_t| \rangle$, then we have $\Delta\mathcal{L}_{w_{t+1}^{\mathrm{Grams}}, w_t} \leq 0$. $\square$

**Theorem D.3** (Loss Descent Comparison, formal version of Theorem 5.4). *Suppose that $\mathcal{L} : \mathbb{R}^d \to \mathbb{R}$ is $L$-smooth. For any parameter vector $w$ at optimization step $t$, let $w_t^{\mathrm{Grams}}$ and $w_t^{\mathrm{C}}$ be the update of Grams in Definition 5.1 and Cautious optimizers in Definition 3.2, respectively. If the stepsize $\eta_t$ satisfies*

$$\eta_t \leq \frac{2}{L\|u_t\|^2} \cdot \min\{\langle u_t \circ g_t, \mathbf{1}_{u_t \circ g_t \geq 0} \rangle, \langle u_t \circ g_t, \mathbf{1}_{u_t \circ g_t < 0} \rangle\},$$

*then we have*

$$\Delta\mathcal{L}_{w_{t+1}^{\mathrm{Grams}}, w_t} \leq \Delta\mathcal{L}_{w_{t+1}^{\mathrm{C}}, w_t} \leq 0.$$

*Proof.* We define the index sets:

$$I^+ = \{i \in [d] : u_{t,i}, g_{t,i} \geq 0\};$$

$$I^- = \{i \in [d] : u_{t,i}, g_{t,i} < 0\}.$$

By Part 1. of Lemma D.2, we have

$$\Delta\mathcal{L}_{w_{t+1}^{\mathrm{Grams}}, w_t} \leq -\eta_t \langle |g_t|, |u_t| \rangle + \frac{L\eta_t^2}{2}\|u_t\|_2^2. \tag{13}$$

By Part 2. of Lemma D.1, we have

$$\Delta\mathcal{L}_{w_{t+1}^{\mathrm{C}},w_t} \geq -\eta_t\langle u_t \circ g_t, \mathbf{1}_{u_t \circ g_t \geq 0}\rangle. \tag{14}$$

Then we can show that

$$\Delta\mathcal{L}_{w_{t+1}^{\mathrm{Grams}},w_t} - \Delta\mathcal{L}_{w_{t+1}^{\mathrm{C}},w_t} \leq -\eta_t\langle|g_t|, |u_t|\rangle + \eta_t\langle u_t \circ g_t, \mathbf{1}_{u_t \circ g_t \geq 0}\rangle + \frac{L\eta_t^2}{2}\|u_t\|_2^2$$

$$= -\eta_t\sum_{i=1}^d |u_{t,i}||g_{t,i}| + \eta_t\sum_{i\in I^+} u_{t,i}g_{t,i} + \frac{L\eta_t^2}{2}\|u_t\|_2^2$$

$$= -\eta_t\sum_{i\in I^+} |u_{t,i}||g_{t,i}| - \eta_t\sum_{i\in I^-} |u_{t,i}||g_{t,i}| + \eta_t\sum_{i\in I^+} u_{t,i}g_{t,i} + \frac{L\eta_t^2}{2}\|u_t\|_2^2$$

$$= -\eta_t\sum_{i\in I^+} u_{t,i}g_{t,i} - \eta_t\sum_{i\in I^-} |u_{t,i}||g_{t,i}| + \eta_t\sum_{i\in I^+} u_{t,i}g_{t,i} + \frac{L\eta_t^2}{2}\|u_t\|_2^2$$

$$= -\eta_t\sum_{i\in I^-} |u_{t,i}||g_{t,i}| + \frac{L\eta_t^2}{2}\|u_t\|_2^2$$

where the first step follows from Eq. (14) and Eq. (13), the second step expands vectors element-wise, the third step follows from that $[d]$ is the disjoint union of $I^+$ and $I^-$, the fourth step follows from that $|u_{t,i}||g_{t,i}| = u_{t,i}g_{t,i}$ for $i \in I^+$, and the last step follows from basic algebra.

To ensure $\Delta\mathcal{L}_{w_{t+1}^{\mathrm{Grams}},w_t} - \Delta\mathcal{L}_{w_{t+1}^{\mathrm{C}},w_t} \leq 0$, it suffices to have

$$-\eta_t\sum_{i\in I^-} |u_{t,i}||g_{t,i}| + \frac{L\eta_t^2}{2}\|u_t\|_2^2 \leq 0.$$

Rearranging the above inequality gives

$$\eta_t \leq \frac{2}{L\|u_t\|_2^2}\sum_{i\in I^-} |u_{t,i}||g_{t,i}|$$

$$= \frac{2}{L\|u_t\|_2^2}\langle g_t \circ u_t, \mathbf{1}_{u_t \circ g_t < 0}\rangle,$$

where the last step follows from the definition of $I^-$ and basic algebra.

Note that by Part 3 of Lemma D.1, if $\eta_t \leq \frac{2}{L\|u_t\|_2^2}\langle g_t \circ u_t, \mathbf{1}_{g_t \circ u_t \geq 0}\rangle$, we have $\mathcal{L}_{w_{t+1}^{\mathrm{C}},w_t} \leq 0$. $\qquad\square$

# E HAMILTONIAN DYNAMICS

**Definition E.1** (Section 2.1 from (Liang et al., 2024)). *Momentum-based algorithms can be typically viewed as monotonic descending algorithms on an augmented loss $H(W, S)$, which satisfies $\min_S H(W, S) = \mathcal{L}(W)$, so that minimizing $\mathcal{L}(W)$ is equivalent to minimizing $H(W, S)$. A typical choice is*

$$H(w, s) = \mathcal{L}(w) + \mathcal{K}(s),$$

*where $\mathcal{K}(\cdot)$ is any lower bounded function. The continuous-time form of most momentum-based algorithms can be written into a Hamiltonian descent form:*

$$\frac{\mathrm{d}}{\mathrm{d}t}w_t = -\nabla\mathcal{K}(s_t) - \Phi_t(\nabla\mathcal{L}(w_t))$$

$$\frac{\mathrm{d}}{\mathrm{d}t}s_t = \nabla\mathcal{L}(w_t) - \Psi_t(\nabla\mathcal{K}(s_t)) \tag{15}$$

*where $H(W, S)$ is a Hamiltonian (or Lyapunov) function that satisfies*

$$\min_S H(W, S) = \mathcal{L}(W), \quad \forall W,$$

*so that minimizing $\mathcal{L}(W)$ reduces to minimizing $H(W, S)$; and $\Phi(\cdot)$, $\Psi(\cdot)$ are two monotonic mappings satisfying*

$$\langle x, \Phi(x) \rangle \geq 0, \qquad\qquad \langle x, \Psi(x) \rangle \geq 0, \qquad\qquad \forall x \in X.$$

*With $\Phi(X) = \Psi(X) = 0$, the system in (15) reduces to the standard Hamiltonian system that keeps $H(W_t, S_t) = const$ along the trajectory. When adding the descending components with $\Phi$ and $\Psi$, the system then keeps $H(W, S)$ monotonically decreasing:*

$$\frac{\mathrm{d}}{\mathrm{d}t} H(w_t, s_t) = -\Delta_H(w_t, s_t) \leq 0,$$

*where*

$$\Delta_H(w_t, s_t) := \langle \nabla \mathcal{L}(w_t), \Phi(\nabla \mathcal{L}(w_t)) \rangle + \langle \mathcal{K}(s_t), \Psi(\mathcal{K}(s_t)) \rangle. \tag{16}$$

*On the other hand, $\mathcal{L}(w)$, which is the true objective, is not necessarily decreasing monotonically.*

$$\frac{\mathrm{d}}{\mathrm{d}t} \mathcal{L}(w_t) = -\Delta_{\mathcal{L}}(w_t, s_t),$$

*where*

$$\Delta_{\mathcal{L}}(w_t, s_t) := \langle \nabla \mathcal{L}(w_t), \nabla \mathcal{K}(s_t) \rangle + \langle \nabla \mathcal{L}(w_t), \Phi_t(\nabla \mathcal{L}(w_t)) \rangle. \tag{17}$$

**Theorem E.2** (Theorem 2.3 in (Liang et al., 2024)). *For Hamiltonian dynamics of Cautious optimizer (in Definition 3.2), we have:*

$$\Delta_H^C(w_t, s_t) := \frac{\mathrm{d}}{\mathrm{d}t} H(w_t, s_t) = \langle x_t, \mathbf{1} - \mathbf{1}_{x_t \geq 0} \rangle - \Delta_H(w_t, s_t).$$

$$\Delta_{\mathcal{L}}^C(w_t) := \frac{\mathrm{d}}{\mathrm{d}t} \mathcal{L}(w_t) = -\langle x_t, \mathbf{1}_{x_t \geq 0} \rangle - \langle \nabla \mathcal{L}(w_t), \Phi_t(\nabla \mathcal{L}(w_t)) \rangle$$

$$= \langle x_t, \mathbf{1} - \mathbf{1}_{x_t \geq 0} \rangle - \Delta_{\mathcal{L}}(w_t, s_t).$$

*where $\Delta_H(w_t, s_t)$ and $\Delta_{\mathcal{L}}(w_t)$ represent the decreasing rates of $H$ and $\mathcal{L}$ in accordance with the system in Definition E.1.*

*Hence:*

- *If $\langle x_t, (\mathbf{1}_d - \mathrm{sign}(x_t)) \rangle \leq 0$ for any $x \in \mathbb{R}^d$, then both $H$ and $\mathcal{L}$ decrease faster than the original system:*

$$\Delta_H^C(w_t, s_t) \leq -\Delta_H(w_t, s_t) \leq 0,$$
$$\Delta_{\mathcal{L}}^C(w_t) \leq -\Delta_{\mathcal{L}}(w_t, s_t).$$

- *If $\langle x_t, \mathrm{sign}(\nabla \mathcal{L}(w_t)) \rangle \geq 0$ for any $x \in \mathbb{R}^d$, then $\mathcal{L}$ decreases monotonically:*

$$\Delta_{\mathcal{L}}^C(w_t) \leq 0.$$

**Theorem E.3** (Convergence of Grams Hamiltonian Dynamics, formal version of Theorem 5.7). *Following the dynamics in Definition 5.6, we have*

$$\Delta_H^{Grams}(w_t, s_t) := \frac{\mathrm{d}}{\mathrm{d}t} H(w_t, s_t) \leq 0,$$

$$\Delta_{\mathcal{L}}^{Grams}(w_t) := \frac{\mathrm{d}}{\mathrm{d}t} \mathcal{L}(w_t) \leq -\Delta_{\mathcal{L}}(w_t, s_t),$$

*where $\Delta_H(w_t, s_t)$ and $\Delta_{\mathcal{L}}(w_t, s_t)$ represent the decreasing rates of $H$ and $\mathcal{L}$ in accordance with the system in Definition E.1.*

*Proof.* Following the dynamics in Definition 5.6, we can calculate $\Delta_H^{\mathrm{Grams}}(w_t, s_t) := \frac{\mathrm{d}}{\mathrm{d}t} H(w_t, s_t)$:

$$\Delta_H^{\mathrm{Grams}}(w_t, s_t)$$

$$= \langle \nabla \mathcal{L}(w_t), \frac{\mathrm{d}}{\mathrm{d}t} w_t \rangle + \langle \nabla \mathcal{K}(s_t), \frac{\mathrm{d}}{\mathrm{d}t} s_t \rangle$$

$$= \langle \nabla \mathcal{L}(w_t), -\text{sign}(\nabla \mathcal{L}(w_t)) \circ |\nabla \mathcal{K}(s_t)| - \Phi_t(\nabla \mathcal{L}(w_t)) \rangle$$
$$+ \langle \mathcal{K}(s_t), \nabla \mathcal{L}(w_t) - \Psi_t(\nabla \mathcal{K}(s_t)) \rangle$$
$$= \langle \nabla \mathcal{L}(w_t), -\text{sign}(\nabla \mathcal{L}(w_t)) \circ |\nabla \mathcal{K}(s_t)| \rangle + \langle \nabla \mathcal{K}(s_t), \nabla \mathcal{L}(w_t) \rangle - \langle \nabla \mathcal{L}(w_t), \Phi_t(\nabla \mathcal{L}(w_t)) \rangle$$
$$- \langle \nabla \mathcal{K}(s_t), \Psi_t(\nabla \mathcal{K}(s_t)) \rangle$$
$$= \langle \nabla \mathcal{L}(w_t), \nabla \mathcal{K}(s_t) \rangle - \langle |\nabla \mathcal{L}(w_t)|, |\nabla \mathcal{K}(s_t)| \rangle - \Delta_H(w_t, s_t)$$
$$\leq 0,$$

where the first step follows from the chain rule for the time derivative of the Hamiltonian $H$, the second step substitutes the dynamics from Definition 5.6, the third step separates the inner products for clearer analysis, the fourth step follows the definition of $\Delta H(w_t, s_t)$ and Fact C.2, and the last step follows Fact C.3, and $-\Delta_H(w_t, s_t) \leq 0$.

Then, we calculate $\Delta_{\mathcal{L}}^{\text{Grams}}(w_t) := \frac{\mathrm{d}}{\mathrm{d}t} \mathcal{L}(w_t)$:

$$\Delta_{\mathcal{L}}^{\text{Grams}}(w_t) = \langle \nabla \mathcal{L}(w_t), -\text{sign}(\nabla \mathcal{L}(w_t)) \circ |\nabla \mathcal{K}(s_t)| - \Phi_t(\nabla \mathcal{L}(w_t)) \rangle$$
$$= \langle \nabla \mathcal{L}(w_t), -\text{sign}(\nabla \mathcal{L}(w_t)) \circ |\nabla \mathcal{K}(s_t)| \rangle - \langle \nabla \mathcal{L}(w_t), \Phi_t(\nabla \mathcal{L}(w_t)) \rangle$$
$$= -\langle |\nabla \mathcal{L}(w_t)|, |\nabla \mathcal{K}(s_t)| \rangle - \langle \nabla \mathcal{L}(w_t), \Phi_t(\nabla \mathcal{L}(w_t)) \rangle$$
$$= \langle \nabla \mathcal{L}(w_t), \nabla \mathcal{K}(s_t) \rangle - \langle |\nabla \mathcal{L}(w_t)|, |\nabla \mathcal{K}(s_t)| \rangle$$
$$- (\langle \nabla \mathcal{L}(w_t), \Phi_t(\nabla \mathcal{L}(w_t)) \rangle + \langle \nabla \mathcal{L}(w_t), \nabla \mathcal{K}(s_t) \rangle)$$
$$= \langle \nabla \mathcal{L}(w_t), \nabla \mathcal{K}(s_t) \rangle - \langle |\nabla \mathcal{L}(w_t)|, |\nabla \mathcal{K}(s_t)| \rangle - \Delta_{\mathcal{L}}(w_t, s_t)$$

where the first step follows from the chain rule, and the second step separates the inner products. The third step follows Fact C.2, the fourth step adds and subtracts the term $\langle \nabla \mathcal{L}(w_t), \nabla \mathcal{K}(s_t) \rangle$ simultaneously, the fifth step follows the definition of $\Delta_{\mathcal{L}}(w_t, s_t)$ from Eq. (2).

Since we know $\langle \nabla \mathcal{L}(w_t), \nabla \mathcal{K}(s_t) \rangle - \langle |\nabla \mathcal{L}(w_t)|, |\nabla \mathcal{K}(s_t)| \rangle \leq 0$ from Fact C.3,

$$\langle \nabla \mathcal{L}(w_t), \nabla \mathcal{K}(s_t) \rangle - \langle |\nabla \mathcal{L}(w_t)|, |\nabla \mathcal{K}(s_t)| \rangle \leq -\Delta_{\mathcal{L}}(w_t, s_t)$$

Thus we complete the proof. $\qquad \square$

**Theorem E.4** (Convergence Comparison of Hamiltonian Dynamics between Grams and Cautious Optimizers, formal version of Theorem 5.8). *From Theorem E.3 and E.2, recall $\Delta_{\mathcal{L}}^{Grams}(w_t)$ and $\Delta_{\mathcal{L}}^{C}(w_t)$:*

$$\Delta_{\mathcal{L}}^{Grams}(w_t) \leq \Delta_{\mathcal{L}}^{C}(w_t).$$

*Proof.* We calculate the difference between $\Delta_{\mathcal{L}}^{\text{Grams}}(w_t)$ and $\Delta_{\mathcal{L}}^{\text{C}}(w_t)$:

$$\Delta_{\mathcal{L}}^{\text{Grams}}(w_t) - \Delta_{\mathcal{L}}^{\text{C}}(w_t) = \langle \nabla \mathcal{L}(w_t), \nabla \mathcal{K}(s_t) \rangle - \langle |\nabla \mathcal{L}(w_t)|, |\nabla \mathcal{L}(w_t)| \rangle - \langle x_t, \mathbf{1} - \mathbf{1}_{x_t \geq 0} \rangle,$$

where $x_t = \nabla \mathcal{L}(w_t) \circ \nabla \mathcal{K}(s_t)$.

By applying Fact C.4, we know:

$$\langle \nabla \mathcal{L}(w_t), \nabla \mathcal{K}(s_t) \rangle - \langle |\nabla \mathcal{L}(w_t)|, |\nabla \mathcal{K}(s_t)| \rangle - \langle x_t, \mathbf{1} - \mathbf{1}_{x_t \geq 0} \rangle \leq 0,$$

with equality if all components of $\nabla \mathcal{L}(w_t) \circ \nabla \mathcal{K}(s_t) \geq 0$.

Thus:

$$\Delta_{\mathcal{L}}^{\text{Grams}}(w_t) - \Delta_{\mathcal{L}}^{\text{C}}(w_t) \leq 0,$$

which implies:

$$\Delta_{\mathcal{L}}^{\text{Grams}}(w_t) \leq \Delta_{\mathcal{L}}^{\text{C}}(w_t).$$

Thus we complete the proof. $\qquad \square$

## F  EXPERIMENTS DETAILS

For the Lion and C-Lion optimizers, we set the learning rate to $\frac{1}{10} \times$ Adam learning rate, as recommended in (Chen et al., 2024).

### F.1 PRE-TRAINING

For the pre-training experiments with Llama 3.2 60M (Dubey et al., 2024), we used the first $2,048,000$ rows of training data from the English section of the C4 dataset (Raffel et al., 2020). Due to the limited computing resources, we trained $1,000$ steps using constant with warm-up scheduler, in order to simulate the beginning part of regular pre-training.We used the first $10,000$ rows of validation data from the English section of the C4 dataset for evaluation. Table 3 provides a detailed summary of the hyperparameters employed.

Table 3: Hyperparameters for Llama 3.2 60M pre-training experiments.

| Optimizers | Grams/AdamW/CAdamW | Lion/CLion |
|---|---|---|
| **Training** | | |
| Epoch | 1 | 1 |
| Learning Rate | 6e-3 | 6e-4 |
| Weight Decay | 0.0 | 0.0 |
| Batch Size | 2048 | 2048 |
| Model Precision | BF16 | BF16 |
| Mix Precision | BF16&TF32 | BF16&TF32 |
| Scheduler | Constant with warm-up | Constant with warm-up |
| Warm-up Steps | 50 | 50 |
| Grad Clipping | 1.0 | 1.0 |
| $\beta_1$ | 0.9 | 0.9 |
| $\beta_2$ | 0.95 | 0.95 |
| $\epsilon$ | 1e-6 | 1e-6 |
| Seq-len | 256 | 256 |
| **Evaluating** | | |
| Precision | BF16 | |
| Seq-len | 256 | |

For the computer vision experiments, we used the CIFAR-10 dataset (Krizhevsky, 2009) to train and evaluate the WideResNet-50-2 model (Zagoruyko & Komodakis, 2016). Table 4 outlines the corresponding hyperparameters.

Table 4: Hyperparameters for WideResNet-50-2 pre-training experiments.

| Optimizers | Grams/AdamW/CAdamW | Lion/CLion |
|---|---|---|
| **Training** | | |
| Epoch | 10 | 10 |
| Learning Rate | 2e-3 | 2e-4 |
| Weight Decay | 0.0 | 0.0 |
| Batch Size | 128 | 128 |
| Model Precision | FP32 | FP32 |
| Mix Precision | None | None |
| Scheduler | Linear | Linear |
| Warm-up Steps | 100 | 100 |
| Grad Clipping | 1.0 | 1.0 |
| $\beta_1$ | 0.9 | 0.9 |
| $\beta_2$ | 0.999 | 0.99 |
| $\epsilon$ | 1e-6 | 1e-6 |
| **Evaluating** | | |
| Precision | FP32 | |

## F.2 FINE-TUNING

For fine-tuning experiments of the Llama 3.2 1B model, Table 5 provides the detailed hyperparameters.

Table 5: Hyperparameters for Llama 3.2 1B fine-tuning experiments.

| Optimizers | Grams/AdamW/CAdamW |
|---|---|
| **Training** | |
| Epoch | 1 |
| Learning Rate | 1e-4 |
| Weight Decay | 0.0 |
| Batch Size | 64 |
| Model Precision | BF16 |
| Mix Precision | BF16&TF32 |
| Scheduler | Cosine |
| Warm-up Ratio | 0.03 |
| Grad Clipping | 1.0 |
| $\beta_1$ | 0.9 |
| $\beta_2$ | 0.999 |
| $\epsilon$ | 1e-6 |
| Seq-len | 512 |
| **Evaluating** | |
| Precision | BF16 |
| Seq-len | 1024 |

For PEFT of the Llama 3.2 3B model, Table 5 provides the detailed hyperparameters.

Table 6: Hyperparameters for Llama 3.2 3B PEFT experiments.

| Optimizers | Grams/AdamW/CAdamW |
|---|---|
| **Training** | |
| Epoch | 1 |
| Learning Rate | 1e-4 |
| Weight Decay | 0.0 |
| Batch Size | 128 |
| Model Precision | BF16 |
| Mix Precision | BF16&TF32 |
| Scheduler | Cosine |
| Warm-up Ratio | 0.03 |
| Grad Clipping | 1.0 |
| $\beta_1$ | 0.9 |
| $\beta_2$ | 0.999 |
| $\epsilon$ | 1e-6 |
| Seq-len | 512 |
| Rank | 128 |
| SORSA (Cao, 2024) $\gamma$ | 1e-3 |
| **Evaluating** | |
| Precision | BF16 |
| Seq-len | 2048 |

## G  IMPACT STATEMENT

Grams advances optimization efficiency by decoupling update direction and magnitude, offering faster and more stable convergence. This has the potential to significantly reduce training costs and energy consumption for large-scale models, making deep learning more accessible and sustainable. We do not foresee any negative potential societal impact of this work.

## LLM USAGE DISCLOSURE

LLMs were used only to polish language, such as grammar and wording. These models did not contribute to idea creation or writing, and the authors take full responsibility for this paper's content.

