# OpenReview forum: "Decoupling Sign and Magnitude in Optimization with Adaptive Momentum Scaling"
_ICLR.cc/2026/Conference — Submitted to ICLR 2026_

### Official Review · Reviewer_a2S3 · 2025-10-28

**Soundness:** 2
**Presentation:** 3
**Contribution:** 3
**Rating:** 4
**Confidence:** 4

**Summary:**

The authors present a novel optimizer for ANNs called Gradient Descent with Adaptive Momentum Scaling (Grams).
Grams extends the idea of cautious optimizers such as C-Adam.
While C-Adam simply ignores momentum terms (e.g., from Adam) that are not aligned with the gradient, Grams flips the signs of the corresponding momentum terms to align them with the gradient.
This results in using the sign of the gradient combined with the magnitude of the momentum term.
The authors provide a theoretical analysis demonstrating a faster loss descent for Grams compared to C-Adam.
Experimentally, Grams is evaluated on Llama 60M trained on the C4 dataset, WRN-50-2 trained on CIFAR-10, and for fine-tuning Llama 3.2 1B.

**Strengths:**

Despite minor language errors, the paper is well-written and easy to follow.
The idea behind Grams is well-motivated and intuitive.

**Weaknesses:**

1. I do not see much significance in the presented theoretical analysis. It is quite obvious that, compared to C-Adam, Grams will perform better in reducing the loss for sufficiently small learning rates, as it aligns better with the gradient. For small learning rates, the update step that reduces the loss most will always be the gradient itself, without using any momentum. It would therefore be interesting to also include SGD (with and without momentum; and Adam) in the theoretical analysis.
2. Furthermore, simply finding an optimal loss-descending step is not the main challenge in ANN training. The authors do not provide any analysis of the generalization capabilities of Grams.
3. The chosen setup for the hyperparameter analysis, using a VAE on CIFAR-10, is somewhat odd. It would be much more interesting to see this analysis for the other experiments (e.g., image classification on CIFAR). Additionally, the sampling of $\beta_1$, $\beta_2$, and $\eta$ is very coarse. The benefits of Grams over C-Adam are marginal and likely not significant. Also, if the best performance was found at the lowest tested values of both $\beta_1$ and $\beta_2$ (i.e., $\beta_1=0.9$ and $\beta_2=0.99$), why were even lower values not tested?
4. The choice of hyperparameters for the other experiments is not clearly described, especially regarding the learning rate $\eta$. Was $\eta$ optimized for Grams, Adam, or C-Adam, or how was it selected? Why were the default values of $\beta_1 = 0.9$ and $\beta_2 = 0.999$ not used for the experiments with Llama 3.2 60M (reported in Table 3), as in the other experiments (Tables 4–6)? It would be valuable to see results for different learning rates for all three optimizers (Adam, C-Adam, and Grams) to verify that the improvements of Grams are not merely due to a better-suited learning rate.
5. The experiments with WRN-50-2 on CIFAR-10 are not ideal. Experiments using models such as ViT on ImageNet would be more appropriate; at the very least, CIFAR-100 should be included.
6. No standard deviations or other significance analyses are provided, which makes it difficult to trust the presented results.

Minor:
- Figures 1 and 2 should span the full textwidth.

**Questions:**

see weaknesses

---

### Official Review · Reviewer_EqAs · 2025-10-30

**Soundness:** 2
**Presentation:** 2
**Contribution:** 3
**Rating:** 4
**Confidence:** 4

**Summary:**

This manuscript challenges gradient descent optimization techniques. Decoupling sign and magnitude from gradient and momentum, the authors propose AMS and Grams, which enables improved optimization. Experiments on VAE, CIFAR, and LLAMA demonstrate these improvements.

**Strengths:**

- Theoretical analysis is provided with sufficient proof and demonstrations to improve convergence.
- Advancing gradient descent optimization is expected to broadly contribute to the research community.
- Source code is available, which eases deployment of the AMS optimizer in practice.

**Weaknesses:**

- The choice of experimental baseline looks arbitrary. What is the reason to choose baseline for training VAE on CIFAR-10? Does this have certain characteristics for optimization?
- CIFAR-10 with WRN for 90% accuracy is a weak baseline. LLM training with 60M parameters is also not large-scale enough.
- I encourage the authors to present or visualize qualitative analysis for sign behavior.
- I think Figure 2 is not strictly the Gram’s sensitivity analysis to \beta_1 and \beta_2. The results demonstrate improvements over Adams but are still sensitive to \beta_1 and \beta_2.
- The results were reported for final performance. The faster optimization, such as the comparison of training time, is not explicitly demonstrated with experiments.
- This is not weakness but a question. As far as I know, the study of ADOPT found a problem in stability and updated their algorithm on Nov 22, 2024. They applied auxiliary clipping to address this issue. Please check the recent arXiv version of Adopt as well as their GitHub repository, which reports this problem. Also, I wonder whether the proposed method is safe from this issue.
- Please check the following mathematics.
    - Fact A.2 applies subtraction such as $x_2 - x_1$. If following this, I think we should use subtraction such as $w_{t+1} - w_t$ at Line 1008.
    - For Theorems D.3 and 5.4, the authors write the upper bound of \eta. For the last term of $u_t g_t 1_{u_t g_t < 0}$, however, I think it becomes negative, which leads to a negative learning rate.
- Writing should be improved.
    - “\epsilon_t” → “\eta_t” at Line 9-10 of Algorithm 2.
    - Check the parenthesis of Line 337, which omits “)”
    - “AMS and Grams offers” → “AMS and Grams offer”
    - “g_T” → “g_t” at Line 1060.
    - “L” → “\Delta L” at Line 1115.
    - “the the property” → “the property”
    - “the second step substitute” → “the second step substitutes”
- For the source code, the authors apply +1 to total_mask. I understand this practice is for avoiding division by zero, but it might affect optimization behavior. A smaller choice would be better.

**Questions:**

Please see the weaknesses above. My score is based on the assumption that all typos are corrected in the revised manuscript.

---

### Official Review · Reviewer_nAWt · 2025-11-01

**Soundness:** 3
**Presentation:** 3
**Contribution:** 3
**Rating:** 6
**Confidence:** 3

**Summary:**

Authors introduce Adaptive Momentum Scaling (AMS) separately tracks the sign and magnitude of momentum. Based on this, they derived Grams optimizer which uses current gradients direction and momentum to scale magnitude. The authors prove discrete-time loss-descent bounds. They also present a Hamiltonian-descent interpretation and empirical results where Grams consistently beats Adam/Lion and their cautious variants on VAE/CIFAR, LLaMA pretraining and fine-tuning.

**Strengths:**

- Overall, authors provide AMS as a unified framework that unifies Adam, C-Adam, and Grams by simple hyperparameter choices.

- The authors provide explicit discrete-time descent bounds and a pointwise comparison theorem. They embed Grams in the Hamiltonian-descent framework, giving continuous-time monotonicity statements and providing interpretable dynamics comparisons similar to the cautious/momentum literature.

- Across the reported results on VAE, WideResNet, LLaMA pretraining and fine-tuning, Grams consistently outperform Adam, Lion and cautious variants.

**Weaknesses:**

- The empirical evaluation somewhat seems to be limited in scope and doesn’t compare Grams to other recent optimizers like Shampoo, K-Fac etc that target similar problems. The appendix provides hyperparameter details, but convergence-time metrics (like wall-clock etc) and statistical significance of reported deltas are missing.

- The discrete-time descent guarantees are pointwise one-step comparisons that rely on local, small enough step-size conditions. How these bounds translate to global convergence in realistic, highly nonconvex training? Or cases like heavy sign flips, high-noise stochastic gradients, or sharpness.

- The results highlight that AMS’s benefit depends on specific hyperparameter choices. When and why should one prefer the update λ=−1 (Grams) over an intermediate λ in practice? Is there a way to predict which λ is best given problem geometry?

**Questions:**

Please refer to my comments in Weaknesses section.

---

### Official Review · Reviewer_YVxc · 2025-11-03

**Soundness:** 2
**Presentation:** 3
**Contribution:** 2
**Rating:** 4
**Confidence:** 4

**Summary:**

This paper introduces Gradient Descent with Adaptive Momentum Scaling (Grams), which is an optimization framework that decouples the direction and magnitude of parameter updates by separately tracking the sign and scale of momentum. They prove it can recover Adam and Cautious Adam as special cases through appropriate hyperparameter choices.

They present theoretical results including discrete-time descent analysis and Hamiltonian descent analysis, proving Grams achieves strictly better loss descent than Cautious optimizers.

They present experiment results for pretraining VAE for CIFAR-10 (sweep beta1 and beta2) against Adam and Cautious-Adam, pretraining Llama 60M on 2M sequences of C4 for 1000 steps, pretraining Wide ResNet-50-2 on CIFAR-10, and full fine-tuning Llama 3.2 1B and parameter efficient fine-tuning on Llama 3.2 3B on 100k examples from MetaMathQA. They compare against Adam, Lion, RMSProp and Cautious variants.

**Strengths:**

It is a good idea to test whether decoupling direction and magnitude in momentum yields improvements. The optimizer is easy to implement (it’s similar to Adam with a key modification to decouple the momentum sign). They include theoretical results showing improvements over the Cautious family of optimizers in certain settings. They show that the AMS framework unifies existing optimizers (Adam, Cautious Adam) as special cases with different hyper parameters. They use a diverse set of domains and tasks for the experiments (vision with CIFAR-10, language with C4, math with MetaMathQA / GSM-8K / MATH, and pretraining / fine-tuning).

**Weaknesses:**

The main weakness in the paper is that it’s not convincing that the baselines are well-tuned state-of-the-art settings, and the models trained are very small. In particular, the pretraining language experiment uses Llama 60M trained for only 1000 steps (due to compute limitations), and they do not include hyperparameter tuning for the baselines. They are missing comparisons against many optimizers mentioned as related work, including ADOPT, Schedule Free, SOAP and Shampoo.

I realize it is difficult to obtain state-of-the-art optimizer results with limited compute, but even on relatively small models it might be feasible to present results against nanogpt speedrun, even if you benchmark using number of tokens seen rather than wall clock time. Or there are two recent papers with extensive pretraining benchmarks (Wen et al 2025, Semenov et al 2025) and it might be feasible to compare a new optimizer using their settings so that you don’t need to re-tune all their baselines. See https://arxiv.org/pdf/2509.01440 and https://arxiv.org/abs/2509.02046.

As a minor comment, some of the related work could be improved:
- “Shampoo” is usually not capitalized
- “While traditional gradient descent guarantees a monotonic decrease in objective function values” ?
- “Earlier methods, such as signSGD (Bernstein et al., 2018), explored similar ideas but focused on reducing communication costs in distributed optimization.” This isn’t the best description of signSGD: there is much work explaining why signSGD behaves similarly or differently from Adam outside of any analysis of communication costs and you could summarize that here.
- “Despite its efficiency, signSGD often underperformed in deep learning tasks, such as ConvNet training, where Lion demonstrated superior performance through advanced momentum mechanisms.” This sentence needs a citation (where did experiments showing this appear?).

**Questions:**

Have you tried benchmarking Grams on any of the settings I mentioned above (either nanoGPT or the two pretraining optimizer benchmark papers)?

---

### Official Review · Reviewer_dtsE · 2025-11-05

**Soundness:** 4
**Presentation:** 3
**Contribution:** 3
**Rating:** 6
**Confidence:** 3

**Summary:**

This paper introduces Grams, a novel optimizer that decouples the direction and magnitude of parameter updates by using gradient signs for direction and adaptive momentum for scaling. It unifies and extends existing optimizers like Adam and Cautious Adam under a general Adaptive Momentum Scaling (AMS) framework, offering both theoretical guarantees and empirical improvements in pre-training and fine-tuning large models.

**Strengths:**

1. Provides formal convergence guarantees through discrete-time and Hamiltonian descent analyses.

2. The methods outperforms strong baselines like Adam, Lion, and their cautious variants across vision and language tasks.

3. The authors propese a general framework. AMS unifies multiple optimizers and allows flexible design choices via hyperparameters.

**Weaknesses:**

1. The proposed method does not compare against recent high-performance optimizers like ADOPT or SOAP-Muon.

2. There is no discussion of the runtime or memory costs relative to simpler methods such as Lion or SGD.

3. Empirical evaluation is limited to smaller models (60M–3B params) and datasets (C4, CIFAR-10, GSM-8K), leaving scalability to trillion-parameter regimes unclear.

**Questions:**

Your empirical evaluation is limited to models up to 3 B parameters and modest-sized datasets such as C4 and CIFAR-10; have you experimented with any dense or MoE models that have mare than 10B parameters, and if so, how does Grams behave under the memory-bandwidth and pipeline-parallel constraints that dominate at that scale?

---

### Meta-Review · Area_Chair_u4dv · 2025-12-24

**Summary:**

The majority of the reviewers raised concerns in both the theoretical side and the significance of the empirical performance.  Specifically, the lemmas and the theorems in this work might not rigorously showcase the proposed method’s benefit, as there is no iteration complexity guarantees available nor a concrete class of optimization problems is identified (where the proposed method enjoys provable acceleration guarantees) in the paper.

A reviewer points out that there is no discussion of the runtime or memory costs relative to simpler methods. They are also concerned about the limited scope of the empirical evaluation, as well as the choices of hyperparameter tuning. Another reviewer raises a concern that no standard deviations or other significance analyses are provided, and hence the presented results might not be fully convincing.

Given that the authors did not reply to the reviewers, this paper is put into the category of "reject".

**Reviewer Concerns:**

The authors give up the opportunities to reply to the reviewers' comments, and hence the concerns are remaining.

**Reviewer Scores:**

The authors give up the opportunities to reply to the reviewers' comments, and hence the question is not applicable.

---

### Decision · Program_Chairs · 2026-01-26

Reject